# Hyperbolic Procrustes Analysis Using Riemannian Geometry

**Ya-Wei Eileen Lin**[†]   **Yuval Kluger**[‡§¶]   **Ronen Talmon**[†]

[†]Viterbi Faculty of Electrical and Computer Engineering, Technion
[‡]Program in Applied Mathematics, Yale University
[§]Interdepartmental Program in Computational Biology and Bioinformatics, Yale University
[¶]Department of Pathology, Yale University
{lin.ya-wei@campus, ronen@ee}.technion.ac.il, yuval.kluger@yale.edu

## Abstract

Label-free alignment between datasets collected at different times, locations, or by different instruments is a fundamental scientific task. Hyperbolic spaces have recently provided a fruitful foundation for the development of informative representations of hierarchical data. Here, we take a purely geometric approach for label-free alignment of hierarchical datasets and introduce hyperbolic Procrustes analysis (HPA). HPA consists of new implementations of the three prototypical Procrustes analysis components: translation, scaling, and rotation, based on the Riemannian geometry of the Lorentz model of hyperbolic space. We analyze the proposed components, highlighting their useful properties for alignment. The efficacy of HPA, its theoretical properties, stability and computational efficiency are demonstrated in simulations. In addition, we showcase its performance on three batch correction tasks involving gene expression and mass cytometry data. Specifically, we demonstrate high-quality unsupervised batch effect removal from data acquired at different sites and with different technologies that outperforms recent methods for label-free alignment in hyperbolic spaces.

## 1  Introduction

A key scientific task in modern data analysis is the alignment of data. The need for alignment often arises since data are acquired in multiple domains, under different environmental conditions, using various acquisition equipment, and at different sites. This paper focuses on the problem of label-free alignment of data embedded in hyperbolic spaces. Recently, hyperbolic spaces have accentuated in geometric representation learning. These non-Euclidean spaces have become popular since they provide a natural embedding of hierarchical data thanks to the exponential growth of the lengths of their geodesic paths [41, 42, 30, 15, 14, 6, 32].

The problem of alignment of data embedded in hyperbolic spaces has been extensively studied, e.g., in the context of natural language processing [49], ontology matching [10], matching two data modalities [40], and improving the embedding in hyperbolic spaces [2]. A few of these studies are based on optimal transport (OT) [2, 22], a classical problem in mathematics [38] that has recently reemerged in modern data analysis, e.g., for domain adaptation [7]. Despite its increasing usage, OT for unsupervised alignment is fundamentally limited [54], since OT (as any density matching approach) cannot recover volume-preserving maps [3, 4, 36].

In this paper, we resort to Procrustes analysis (PA) [17, 18] that is based on purely geometric considerations. PA has been widely used for aligning datasets by eliminating the shift, scaling, and rotational factors. Over the years, it has been successfully applied to various applications, e.g.,

35th Conference on Neural Information Processing Systems (NeurIPS 2021).

image registration [34], manifold alignment [52], shape matching [35], domain adaptation [47], and manifold learning [27], to name but a few. Here, we address the problem of label-free matching of hierarchical data embedded in hyperbolic spaces. We present hyperbolic Procrustes analysis (HPA), a new PA method in the Lorentz model of hyperbolic geometry. The main novelty lies in the introduction of new implementations of the three prototypical PA components based on Riemannian geometry. Specifically, translation is viewed as a Riemannian mean alignment, implemented using parallel transport (PT). Scaling is determined with respect to geodesic paths. Rotation is considered as moment alignment on a mapping of the tangent space of the manifold to a Euclidean vector space. Our analysis provides new derivations in the Riemannian geometry of the Lorentz model and specifies the commuting properties of the HPA components. We show that HPA, compared to existing baselines and OT-based methods, achieves improved alignment in a purely unsupervised setting. In addition, it has a natural and stable out-of-sample extension, it supports both small and big data, and it is computationally efficient.

We show application to batch correction in bioinformatics tasks. We present results on both gene expression and mass cytometry (CyTOF) data, exemplifying the generality and broad scope of our method. In contrast to recent works [28, 50], our method does not require landmark correspondence, which is often unavailable in many datasets or hard to obtain. Specifically, we show that batch effects caused by acquisition using different technologies, at different sites, and at different times can be accurately removed, while preserving the intrinsic structure of the data.

Our main contributions are as follows. (i) We present a new implementation of PA using the Riemannian geometry of the Lorentz model for unsupervised label-free hierarchical data alignment. (ii) We provide theoretical analysis and justification of our alignment method based on new derivations of Riemannian geometry operations in the Lorentz model. These derivations have their own merit as they could be used in other contexts. (iii) We show experimental results of accurate batch effect removal from several hierarchical bioinformatics datasets without landmark correspondence.

## 2  Background on hyperbolic geometry

Hyperbolic space is a non-Euclidean space with a negative constant sectional curvature and an underlying geometry that describes tree-like graphs with small distortions [46]. There exist four commonly-used models for hyperbolic spaces: Poincaré disk model, Lorentz model (hyperboloid model), Poincaré half-plane model, and Beltrami-Klein model. These four models are equivalent and there exist transformations between them. Here, we consider the Lorentz model, and specifically, the upper sheet of the hyperboloid model, because its basic Riemannian operations have simple closed-form expressions and the computation of the geodesic distances is stable [42, 30].

Formally, the upper sheet of the hyperboloid model in a $d$-dimensional hyperbolic space is defined by $\mathbb{L}^d := \{ \boldsymbol{x} \in \mathbb{R}^{d+1} | \langle \boldsymbol{x}, \boldsymbol{x} \rangle_{\mathcal{L}} = -1, \boldsymbol{x}(1) > 0 \}$, where $\langle \boldsymbol{x}, \boldsymbol{x} \rangle_{\mathcal{L}} = \boldsymbol{x}^{\top} \boldsymbol{H} \boldsymbol{x}$ is the Lorentzian inner product and $\boldsymbol{H} \in \mathbb{R}^{(d+1) \times (d+1)}$ is defined by $\boldsymbol{H} = [-1, \boldsymbol{0}^{\top}; \boldsymbol{0}, \boldsymbol{I}_d]$. The Lorentzian norm of a hyperbolic vector $\boldsymbol{x} \in \mathbb{L}^d$ is denoted by $||\boldsymbol{x}||_{\mathcal{L}} = \sqrt{\langle \boldsymbol{x}, \boldsymbol{x} \rangle_{\mathcal{L}}}$, with the origin $\boldsymbol{\mu}_0 = [1, \boldsymbol{0}^{\top}]^{\top} \in \mathbb{L}^d$. Let $\mathcal{T}_{\boldsymbol{x}} \mathbb{L}^d$ be the tangent space at $\boldsymbol{x} \in \mathbb{L}^d$, defined by $\mathcal{T}_{\boldsymbol{x}} \mathbb{L}^d := \{ \boldsymbol{v} | \langle \boldsymbol{x}, \boldsymbol{v} \rangle_{\mathcal{L}} = 0 \}$. Consider $\boldsymbol{x} \in \mathbb{L}^d$ and $\boldsymbol{v} \in \mathcal{T}_{\boldsymbol{x}} \mathbb{L}^d$, the geodesic path $\phi : \mathbb{R}_0^+ \to \mathbb{L}^d$ is defined by $\phi(t) = \cosh(||\boldsymbol{v}||_{\mathcal{L}} t) \boldsymbol{x} + \sinh(||\boldsymbol{v}||_{\mathcal{L}} t) \frac{\boldsymbol{v}}{||\boldsymbol{v}||_{\mathcal{L}}}$ with $\phi(0) = \boldsymbol{x}$ and initial velocity $\phi'(0) = \boldsymbol{v}$, where $\phi'(t) := \frac{d}{dt} \phi(t)$. In addition, the associated geodesic distance is $d_{\mathbb{L}^d}(\boldsymbol{x}, \phi_{\boldsymbol{v}}(t)) = \cosh^{-1}(-\langle \boldsymbol{x}, \phi_{\boldsymbol{v}}(t) \rangle_{\mathcal{L}})$.

The Exponential map, projecting a point $\boldsymbol{v} \in \mathcal{T}_{\boldsymbol{x}} \mathbb{L}^d$ to the manifold $\mathbb{L}^d$, is given by $\mathrm{Exp}_{\boldsymbol{x}}(\boldsymbol{v}) = \phi(1) = \cosh(||\boldsymbol{v}||_{\mathcal{L}}) \boldsymbol{x} + \sinh(||\boldsymbol{v}||_{\mathcal{L}}) \frac{\boldsymbol{v}}{||\boldsymbol{v}||_{\mathcal{L}}}$. The Logarithmic map, projecting a point $\boldsymbol{y} \in \mathbb{L}^d$ to the tangent space $\mathcal{T}_{\boldsymbol{x}} \mathbb{L}^d$ at $\boldsymbol{x}$, is defined by $\mathrm{Log}_{\boldsymbol{x}}(\boldsymbol{y}) = \frac{\cosh^{-1}(\lambda)}{\sqrt{\lambda^2 - 1}} (\boldsymbol{y} - \lambda \boldsymbol{x})$, where $\lambda = -\langle \boldsymbol{x}, \boldsymbol{y} \rangle_{\mathcal{L}}$. The PT of a vector $\boldsymbol{v} \in \mathcal{T}_{\boldsymbol{x}} \mathbb{L}^d$ along the geodesic path from $\boldsymbol{x} \in \mathbb{L}^d$ to $\boldsymbol{y} \in \mathbb{L}^d$ is defined by $\mathrm{PT}_{\boldsymbol{x} \to \boldsymbol{y}}(\boldsymbol{v}) = \boldsymbol{v} + \frac{\langle \boldsymbol{y} - \lambda \boldsymbol{x}, \boldsymbol{v} \rangle_{\mathcal{L}}}{\lambda + 1}(\boldsymbol{x} + \boldsymbol{y})$, where $\lambda = -\langle \boldsymbol{x}, \boldsymbol{y} \rangle_{\mathcal{L}}$, keeping the metric tensor unchanged. The Riemannian mean $\overline{\boldsymbol{x}}_{\mathcal{X}}$ and the corresponding dispersion $d_{\mathcal{X}}$ of a set $\mathcal{X} = \{ \boldsymbol{x}_i | \boldsymbol{x}_i \in \mathbb{L}^d \}_{i=1}^n$ are defined using the Fréchet mean [13, 33] by

$$\overline{\boldsymbol{x}}_{\mathcal{X}} := m(\mathcal{X}) = \underset{\boldsymbol{x} \in \mathbb{L}^d}{\arg\min} \sum_{i=1}^n d_{\mathbb{L}^d}^2(\boldsymbol{x}, \boldsymbol{x}_i) \text{ and } d_{\mathcal{X}} := r(\mathcal{X}) = \underset{\boldsymbol{x} \in \mathbb{L}^d}{\min} \frac{1}{n} \sum_{i=1}^n d_{\mathbb{L}^d}^2(\boldsymbol{x}, \boldsymbol{x}_i), \quad (1)$$

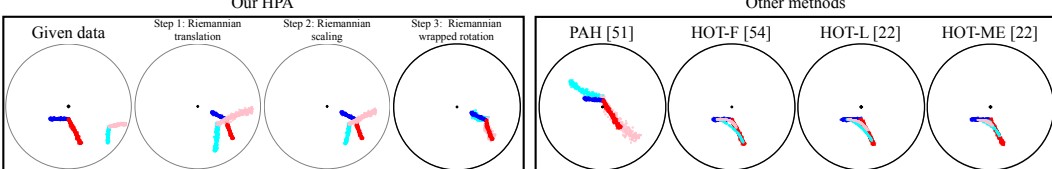

Figure 1: Illustration of our alignment method (HPA). Two sets of points in hyperbolic space are given, depicted in dark and light colors. Each point is associated with one of two labels: (i) blue/cyan, and (ii) red/pink. Left: Alignment results after each step of our HPA implemented in $\mathbb{L}^2$: Riemannian translation, Riemannian scaling, and Riemannian wrapped rotation. Note that these Riemannian operations in hyperbolic space are different than their Euclidean counterparts. Right: Alignment results of four methods: PAH [51] and HOT-F [54] applied in $\mathbb{L}^2$, and HOT-L [22] and HOT-ME [22] applied in the 2D Poincaré disk. For visualization, all points in $\mathbb{L}^2$ are transformed to the 2D Poincaré disk. The alignment after Step 3 of HPA (circled in black) is more accurate than the alignments obtained by the other methods.

where $m : \mathcal{X} \to \mathbb{L}^d$ and $r : \mathcal{X} \to \mathbb{R}^+$. Note that the Fréchet mean of samples on connected and compact Riemannian manifolds of non-positive curvatures, such as hyperbolic spaces, is guaranteed to exist, and it is unique [24, 44, 1]. The Fréchet mean is commonly computed by the Karcher Flow [24, 20], which is computationally demanding. Importantly, in the considered hyperbolic space, the Fréchet mean can be efficiently obtained using the accurate gradient formulation [33].

Given a vector $\overline{x} \in \mathbb{L}^d$ and a symmetric and positive-definite (SPD) matrix $\Sigma \in \mathbb{R}^{d \times d}$, the wrapped normal distribution $\mathcal{G}(\overline{x}, \Sigma)$ provides a generative model of hyperbolic samples as follows [39, 11]. First, a vector $v'$ is sampled from $\mathcal{N}(\mathbf{0}, \Sigma)$. Then, $0$ is concatenated to the vector $v'$ such that $v = [0, v']^\top \in \mathcal{T}_{\mu_0} \mathbb{L}^d$. Finally, PT from the origin $\mu_0 = [1, \mathbf{0}^\top]^\top$ to $\overline{x}$ is applied to $v$, and the resulting point is mapped to the manifold using the Exponential map at $\overline{x}$. The probability density function of this model is given by $\log \mathcal{G}(y|\overline{x}, \Sigma) = \log \mathcal{N}(\mathbf{0}, \Sigma) - (n-1)\log(\frac{\sinh \|v'\|_2}{\|v'\|_2})$.

## 3 Hyperbolic Procrustes analysis

Existing methods for data alignment typically seek a function that minimizes a certain cost. A large body of work attempts to match the empirical densities of two datasets, e.g., by minimizing the maximum mean discrepancy (MMD) [48, 29] or solving OT problems [45, 2, 22]. Finding an effective cost function without labels or landmarks is challenging, and minimizing such costs directly often lead to poor alignment in practice (see illustration in Fig. 1). A different well-established approach that applies indirect alignment based on geometric considerations is PA. While preparing this manuscript, another method of PA in hyperbolic spaces (PAH) was presented for matching two sets, assuming that they consist of the same number of points and that there exists a point-wise isometric map between them [51]. We remark that the analysis we present here applies to broader settings and makes no such assumptions. See Appendix E for details on classical PA as well as for comparisons to [51] and to the application of Euclidean PA in the tangent space.

We consider two sets of points $\mathcal{H}^{(1)} = \{h_i^{(1)}\}_{i=1}^{N_1}$ and $\mathcal{H}^{(2)} = \{h_i^{(2)}\}_{i=1}^{N_2}$ in $\mathbb{L}^d$. Here, we aim to find a function $\zeta : \mathbb{L}^d \to \mathbb{L}^d$, consisting of three components: translation, scaling, and rotation, that aligns $\mathcal{H}^{(2)}$ with $\mathcal{H}^{(1)}$ in an unsupervised label-free manner as depicted in Fig. 1. Finding such a function can be viewed as an extension of classical PA from the Euclidean space $\mathbb{R}^{d+1}$ to the Lorentz model $\mathbb{L}^d$. A natural extension to multiple sets is described in Section 3.5. We remark that the statements are written in the context of the problem at hand. In Appendix A, we restate them more generally and present their proofs.

### 3.1 Riemannian translation

Let $\overline{h}^{(1)}$ and $\overline{h}^{(2)}$ denote the Riemannian means of the sets $\mathcal{H}^{(1)}$ and $\mathcal{H}^{(2)}$, respectively. In this translation component, we find a map $\Gamma_{\overline{h}^{(2)} \to \overline{h}^{(1)}} : \mathbb{L}^d \to \mathbb{L}^d$ that aligns the Riemannian means of

the sets. In the spirit of [5, 53], we propose to construct $\Gamma_{\overline{h}^{(2)} \to \overline{h}^{(1)}}(h_i^{(2)})$ as the composition of three Riemannian operations in $\mathbb{L}^d$: the Logarithmic map applied to $h_i^{(2)}$ at $\overline{h}^{(2)}$, PT $h_i^{(2)}$ from $\overline{h}^{(2)}$ to $\overline{h}^{(1)}$ along the geodesic path, and the Exponential map applied to the transported point at $\overline{h}^{(1)}$:

$$\Gamma_{\overline{h}^{(2)} \to \overline{h}^{(1)}}(h_i^{(2)}) := \mathrm{Exp}_{\overline{h}^{(1)}}(\mathrm{PT}_{\overline{h}^{(2)} \to \overline{h}^{(1)}}(\mathrm{Log}_{\overline{h}^{(2)}}(h_i^{(2)}))). \tag{2}$$

See Fig. B.1 in Appendix B for illustration. Since the geodesic path between any two points in $\mathbb{L}^d$ is unique [46], $\Gamma_{\overline{h}^{(2)} \to \overline{h}^{(1)}}$ is well-defined. The rationale behind the combination of these three Riemannian operations is twofold. First, PT is a map that aligns the means of the sets, while preserving their internal structure. Second, the Logarithmic and Exponential maps compose a map whose domain and range are the Lorentz model $\mathbb{L}^d$ rather than the tangent space, as desired. We make these claims formal in the following results.

**Proposition 1.** *The map* $\Gamma_{\overline{h}^{(2)} \to \overline{h}^{(1)}}$ *defined in Eq.* (2) *aligns the means of the sets, i.e., it satisfies*

$$\overline{h}^{(1)} = m(\{\Gamma_{\overline{h}^{(2)} \to \overline{h}^{(1)}}(h_i^{(2)})\}_{i=1}^{N_2}), \tag{3}$$

*where* $m$ *is the function defined in Eq.* (1).

**Proposition 2.** *The map* $\Gamma_{\overline{h}^{(2)} \to \overline{h}^{(1)}}(h_i^{(2)})$ *for all* $h_i^{(2)} \in \mathcal{H}^{(2)}$ *can be recast as:*

$$\Gamma_{\overline{h}^{(2)} \to \overline{h}^{(1)}}(h_i^{(2)}) = h_i^{(2)} - \beta(h_i^{(2)}|\overline{h}^{(1)}, \overline{h}^{(2)})\overline{h}^{(2)} + \gamma(h_i^{(2)}|\overline{h}^{(1)}, \overline{h}^{(2)})\overline{h}^{(1)}, \tag{4}$$

*where the functions* $\beta$ *and* $\gamma$ *are positive, defined by* $0 < \beta(h_i^{(2)}|\overline{h}^{(1)}, \overline{h}^{(2)}) = -\left\langle \frac{\overline{h}^{(1)}+\overline{h}^{(2)}}{\alpha+1}, h_i^{(2)} \right\rangle_{\mathcal{L}}$ *and* $0 < \gamma(h_i^{(2)}|\overline{h}^{(1)}, \overline{h}^{(2)}) = \left\langle \frac{\overline{h}^{(1)}-(2\alpha+1)\overline{h}^{(2)}}{\alpha+1}, h_i^{(2)} \right\rangle_{\mathcal{L}}$, *respectively, and* $0 < \alpha = -\langle \overline{h}^{(1)}, \overline{h}^{(2)} \rangle_{\mathcal{L}}$.

In addition to providing a compact closed-form expression, Prop. 2 gives the proposed translation based on Riemannian geometry an interpretation of standard mean alignment in linear vector spaces. It implies that the alignment is nothing but subtracting the mean of the source set $\overline{h}^{(2)}$ from each vector in $\mathcal{H}^{(2)}$, and adding the mean of the target set $\overline{h}^{(1)}$ (with the appropriate scales).

**Proposition 3.** *The map* $\Gamma_{\overline{h}^{(2)} \to \overline{h}^{(1)}}$ *preserves distances (i.e., it is an isometry):*

$$d_{\mathbb{L}^d}(h_i^{(2)}, h_j^{(2)}) = d_{\mathbb{L}^d}(\Gamma_{\overline{h}^{(2)} \to \overline{h}^{(1)}}(h_i^{(2)}), \Gamma_{\overline{h}^{(2)} \to \overline{h}^{(1)}}(h_j^{(2)})), \tag{5}$$

*for any two points* $h_i^{(2)}, h_j^{(2)} \in \mathcal{H}^{(2)}$.

Let $\psi(t)$ be the unique geodesic path from $\overline{h}^{(2)}$ to $\overline{h}^{(1)}$ such that $\psi(0) = \overline{h}^{(2)}$ and $\psi(1) = \overline{h}^{(1)}$, and let $\psi'(0) \in \mathcal{T}_{\overline{h}^{(2)}}\mathbb{L}^d$ and $\psi'(1) \in \mathcal{T}_{\overline{h}^{(1)}}\mathbb{L}^d$ be the corresponding velocities, respectively.

**Proposition 4.** *The map* $\Gamma_{\overline{h}^{(2)} \to \overline{h}^{(1)}}$ *aligns geodesic velocities, i.e., given the mapping of the geodesic velocities to the manifold* $\mathbb{L}^d \ni v_0 = \mathrm{Exp}_{\overline{h}^{(2)}}(\psi'(0)) = \overline{h}^{(1)}$ *and* $\mathbb{L}^d \ni v_1 = \mathrm{Exp}_{\overline{h}^{(1)}}(\psi'(1))$, *we have*

$$\Gamma_{\overline{h}^{(2)} \to \overline{h}^{(1)}}(v_0) = v_1. \tag{6}$$

Isometry is determined up to rotation, a fact that can be problematic for alignment. For example, any H-unitary matrix [16] can be an isometric function in $\mathbb{L}^d$. When landmarks are given, they can be used to alleviate this redundancy. However, in the purely unsupervised setting we consider, other data-driven ques are required. Prop. 4 implies that the proposed translation based on PT fixes some of these rotational degrees of freedom by aligning the geodesic velocities. In Section 3.3 we revisit this issue. Now, with a slight abuse of notation, let $\widetilde{\mathcal{H}}^{(2)} = \Gamma_{\overline{h}^{(2)} \to \overline{h}^{(1)}}(\mathcal{H}^{(2)})$.

**Proposition 5.** *Consider two subsets* $\mathcal{A}, \mathcal{B} \subset \mathcal{H}^{(2)}$ *and their translations* $\widetilde{\mathcal{A}} = \Gamma_{\overline{h}^{(2)} \to \overline{h}^{(1)}}(\mathcal{A})$, $\widetilde{\mathcal{B}} = \Gamma_{\overline{h}^{(2)} \to \overline{h}^{(1)}}(\mathcal{B}) \subset \widetilde{\mathcal{H}}^{(2)}$. *Let* $\overline{a} = m(\mathcal{A}), \overline{b} = m(\mathcal{B}), \widetilde{a} = m(\widetilde{\mathcal{A}})$, *and* $\widetilde{b} = m(\widetilde{\mathcal{B}})$ *be the Riemannian means of the subsets. Then,*

$$\Gamma_{\overline{h}^{(2)} \to \overline{h}^{(1)}} \circ \Gamma_{\overline{a} \to \overline{b}} = \Gamma_{\widetilde{a} \to \widetilde{b}} \circ \Gamma_{\overline{h}^{(2)} \to \overline{h}^{(1)}}. \tag{7}$$

In the context of the alignment problem, the importance of Prop. 5 is the following. Suppose the two sets correspond to data measured at two labs (denoted with and without a tilde), and suppose each set was acquired by two types of equipment (denoted by $\mathcal{A}$ and $\mathcal{B}$). Prop. 5 implies that aligning data from the different labs and then aligning data acquired using the different equipment is equivalent to first aligning the different equipment and then the different labs, i.e., any order of the two alignments generates the same result. Seemingly, this is a natural property in Euclidean spaces. However, in a Riemannian manifold, it is not a trivial result, and it holds for the transport along the geodesic path. See Appendix A for counter-examples.

## 3.2 Riemannian scaling

Let $d^{(1)}$ and $d^{(2)}$ denote the Riemannian dispersions of $\mathcal{H}^{(1)}$ and $\mathcal{H}^{(2)}$. By propositions 1 and 3, $\overline{h}^{(1)}$ and $d^{(2)}$ are the mean and dispersion of $\widetilde{\mathcal{H}}^{(2)}$. Here, our goal is to align the Riemannian dispersions of $\mathcal{H}^{(1)}$ and $\widetilde{\mathcal{H}}^{(2)}$. For this purpose, we propose the scaling function $\Upsilon^s_{\overline{h}^{(1)}} : \mathbb{L}^d \to \mathbb{L}^d$, given by

$$\Upsilon^s_{\overline{h}^{(1)}}(\widetilde{h}^{(2)}_i) = \phi_i(s), \tag{8}$$

where $s = \sqrt{d^{(1)}/d^{(2)}}$ is the scaling factor and $\phi_i(t)$ is the geodesic path from $\overline{h}^{(1)}$ to $\widetilde{h}^{(2)}_i$ such that $\phi_i(0) = \overline{h}^{(1)}$ and $\phi_i(1) = \widetilde{h}^{(2)}_i$. See Fig. B.2 in Appendix B for illustration.

**Proposition 6.** *The dispersion of the rescaled set $\widehat{\mathcal{H}}^{(2)} = \Upsilon^s_{\overline{h}^{(1)}}(\widetilde{\mathcal{H}}^{(2)})$ is $d^{(1)}$.*

## 3.3 Riemannian wrapped rotation

The purpose of this component is to align the orientation of the distributions of the two sets after translation and scaling, namely, after aligning their first and second moments. The proposed rotation function $\Theta_{\overline{h}^{(1)}} : \mathbb{L}^d \to \mathbb{L}^d$ consists of (i) mapping the points from the manifold $\mathbb{L}^d$ to the tangent space $\mathcal{T}_{\overline{h}^{(1)}}\mathbb{L}^d$, (ii) mapping to $\mathbb{R}^d$, (iii) rotating in $\mathbb{R}^d$, and (iv) mapping back to the tangent space and then to the manifold. We perform the rotation in $\mathbb{R}^d$, which we term *wrapped rotation*, rather than a direct rotation on the manifold $\mathbb{L}^d$ or on the tangent space $\mathcal{T}_{\overline{h}^{(1)}}\mathbb{L}^d$ for the following reasons.

First, the frequently used rotation map in $\mathbb{L}^d$ [51] does not necessarily preserve the Riemannian mean, and in our context, it might reverse the mean alignment. Second, rotation applied directly to the tangent space $\mathcal{T}_{\overline{h}^{(1)}}\mathbb{L}^d$ does not guarantee that the rotated points remain on the same tangent space. Third, applying rotation in the Euclidean vector space that is isometric to the tangent space is less efficient and stable, and it obtains slightly worse empirical results (see Appendix D.4 for details). Last, applying the rotation in $\mathbb{R}^d$ allows us to use the standard Euclidean rotation using SVD. In Section 4, we empirically demonstrate the advantage of the proposed rotation compared to the alternatives.

**Definition 1.** *Let the mapping function $\mathcal{P}_{\overline{h}^{(1)}} : \mathcal{T}_{\overline{h}^{(1)}}\mathbb{L}^d \to \mathbb{R}^d$ defined on the tangent space at $\overline{h}^{(1)}$ and its inverse map be the following functions defined by*

$$\mathcal{P}_{\overline{h}^{(1)}}(\boldsymbol{v}) := \big[\boldsymbol{v}(2), \ldots, \boldsymbol{v}(d+1)\big]^\top \in \mathbb{R}^d \text{ and } \mathcal{P}^{-1}_{\overline{h}^{(1)}}(\boldsymbol{s}) := \left[\frac{\langle \boldsymbol{s}, \mathcal{P}_{\overline{h}^{(1)}}(\overline{h}^{(1)})\rangle}{\overline{h}^{(1)}(1)}, \boldsymbol{s}^\top\right]^\top \in \mathcal{T}_{\overline{h}^{(1)}}\mathbb{L}^d, \tag{9}$$

*where $\boldsymbol{s} \in \mathbb{R}^d$ and $\langle \cdot, \cdot \rangle$ is the standard Euclidean inner product.*

Note that removing the first element of $\boldsymbol{v}$ is valid due to the constraint imposed on the vector elements in the tangent space by definition. Indeed, no information is lost and the mapping is invertible.

The first step in our rotation component is to map the points in $\mathcal{H}^{(1)}$ and in $\widehat{\mathcal{H}}^{(2)}$ to the tangent space at $\overline{h}^{(1)}$: $\boldsymbol{v}^{(1)}_i = \mathrm{Log}_{\overline{h}^{(1)}}(\boldsymbol{h}^{(1)}_i)$ for $i = 1, \ldots, N_1$, and $\boldsymbol{v}^{(2)}_i = \mathrm{Log}_{\overline{h}^{(1)}}(\widehat{\boldsymbol{h}}^{(2)}_i)$ for $i = 1, \ldots, N_2$. In the second step, we map the points by the mapping function in Definition 1 and re-center them: $\boldsymbol{s}^{(1)}_i = \mathcal{P}_{\overline{h}^{(1)}}(\boldsymbol{v}^{(1)}_i) - \overline{\boldsymbol{s}}^{(1)}$, $i = 1, \ldots, N_1$ and $\boldsymbol{s}^{(2)}_i = \mathcal{P}_{\overline{h}^{(1)}}(\boldsymbol{v}^{(2)}_i) - \overline{\boldsymbol{s}}^{(2)}$, $i = 1, \ldots, N_2$, where $\overline{\boldsymbol{s}}^{(k)} = \frac{1}{N_k} \sum_{i=1}^{N_k} \mathcal{P}_{\overline{h}^{(1)}}(\boldsymbol{v}^{(k)}_i)$ for $k = 1, 2$ is the mean vector of the projections. Then, the mapped and centered points (in $\mathbb{R}^d$) are collected into matrices:

$$\boldsymbol{S}^{(k)} = \big[\boldsymbol{s}^{(k)}_1, \boldsymbol{s}^{(k)}_2, \ldots, \boldsymbol{s}^{(k)}_{N_k}\big] \in \mathbb{R}^{d \times N_k}. \tag{10}$$

In the third step, for each set $k = 1, 2$, we compute the rotation matrix $\boldsymbol{U}^{(k)} \in \mathbb{R}^{d \times d}$ by applying SVD to the matrix $\boldsymbol{S}^{(k)} = \boldsymbol{U}^{(k)} \boldsymbol{\Lambda}^{(k)} (\boldsymbol{E}^{(k)})^{\top}$. Since the left-singular vectors are determined up to a sign, we propose to align their signs as follows: $\boldsymbol{u}_i^{(2)} \leftarrow \mathrm{sign}(\langle \boldsymbol{u}_i^{(2)}, \boldsymbol{u}_i^{(1)} \rangle) \boldsymbol{u}_i^{(2)}$, where $\boldsymbol{u}_i^{(1)}$ and $\boldsymbol{u}_i^{(2)}$ are the $i$-th left-singular vector of the two sets, resulting in modified rotation matrices $\boldsymbol{U}^{(k)}$. Finally, we apply the rotation to $\widehat{\mathcal{H}}^{(2)}$ by

$$\Theta_{\overline{\boldsymbol{h}}^{(1)}}^{\boldsymbol{U}}(\widehat{\boldsymbol{h}}_i^{(2)}) = \mathrm{Exp}_{\overline{\boldsymbol{h}}^{(1)}} \left( \mathcal{P}_{\overline{\boldsymbol{h}}^{(1)}}^{-1} \left( \boldsymbol{U}^{\top} \left( \mathcal{P}_{\overline{\boldsymbol{h}}^{(1)}} (\mathrm{Log}_{\overline{\boldsymbol{h}}^{(1)}}(\widehat{\boldsymbol{h}}_i^{(2)})) - \overline{\boldsymbol{s}}^{(2)} \right) + \overline{\boldsymbol{s}}^{(2)} \right) \right), \quad (11)$$

where $\boldsymbol{U} = \boldsymbol{U}^{(1)} (\boldsymbol{U}^{(2)})^{\top}$.

**Proposition 7.** *The wrapped rotation is bijective, and the inverse is given by*

$$(\Theta_{\overline{\boldsymbol{h}}^{(1)}}^{\boldsymbol{U}})^{-1} = \Theta_{\overline{\boldsymbol{h}}^{(1)}}^{\boldsymbol{U}^{\top}}. \quad (12)$$

### 3.4 Analysis

Putting all three components together, the proposed HPA that aligns $\mathcal{H}^{(2)}$ with $\mathcal{H}^{(1)}$ culminates in the composition of translation, scaling, and rotation:

$$\Theta_{\overline{\boldsymbol{h}}^{(1)}}^{\boldsymbol{U}} \circ \Upsilon_{\overline{\boldsymbol{h}}^{(1)}}^{s} \circ \Gamma_{\overline{\boldsymbol{h}}^{(2)} \to \overline{\boldsymbol{h}}^{(1)}}. \quad (13)$$

As in most PA schemes, the order of the three components is important. Yet, the proposed components allow us a certain degree of freedom, as indicated in the following results.

**Proposition 8.** *The Riemannian translation and the Riemannian scaling commute w.r.t. the Riemannian means $\overline{\boldsymbol{h}}^{(1)}$ and $\overline{\boldsymbol{h}}^{(2)}$:*

$$\Upsilon_{\overline{\boldsymbol{h}}^{(1)}}^{s} \circ \Gamma_{\overline{\boldsymbol{h}}^{(2)} \to \overline{\boldsymbol{h}}^{(1)}} = \Gamma_{\overline{\boldsymbol{h}}^{(2)} \to \overline{\boldsymbol{h}}^{(1)}} \circ \Upsilon_{\overline{\boldsymbol{h}}^{(2)}}^{s}. \quad (14)$$

Note that $\Upsilon_{\overline{\boldsymbol{h}}^{(1)}}^{s}$ and $\Gamma_{\overline{\boldsymbol{h}}^{(2)} \to \overline{\boldsymbol{h}}^{(1)}}$ do not necessarily commute: $\Upsilon_{\overline{\boldsymbol{h}}^{(1)}}^{s} \circ \Gamma_{\overline{\boldsymbol{h}}^{(2)} \to \overline{\boldsymbol{h}}^{(1)}} \neq \Gamma_{\overline{\boldsymbol{h}}^{(2)} \to \overline{\boldsymbol{h}}^{(1)}} \circ \Upsilon_{\overline{\boldsymbol{h}}^{(1)}}^{s}$.

**Proposition 9.** *The Riemannian scaling and the wrapped rotation commute:*

$$\Upsilon_{\overline{\boldsymbol{h}}^{(1)}}^{s} \circ \Theta_{\overline{\boldsymbol{h}}^{(1)}}^{\boldsymbol{U}} = \Theta_{\overline{\boldsymbol{h}}^{(1)}}^{\boldsymbol{U}} \circ \Upsilon_{\overline{\boldsymbol{h}}^{(1)}}^{s}. \quad (15)$$

We note that the rotation does not commute with the translation, because PT only preserves the local covariant derivative on the tangent space but might cause rotation and distortion along the transportation. Therefore, the rotation is required to be the last component of our HPA.

Thus far, we did not present a model for the discrepancy between the two sets, nor we presented the proposed HPA as optimal with respect to some criterion. In the following result, we show that if the discrepancy between the sets can be expressed as a composition of translation, scaling, and rotation, then the two sets can be perfectly aligned using HPA.

**Proposition 10.** *Let $\eta : \mathbb{L}^d \to \mathbb{L}^d$ be a map, given by $\eta = \Theta_{\overline{\boldsymbol{h}}^{(1)}}^{\boldsymbol{U}} \circ \Upsilon_{\overline{\boldsymbol{h}}^{(1)}}^{s} \circ \Gamma_{\overline{\boldsymbol{h}}^{(2)} \to \overline{\boldsymbol{h}}^{(1)}}$. If $\mathcal{H}^{(1)} = \{\boldsymbol{h}_i^{(1)} = \eta(\boldsymbol{h}_i^{(2)})\}_{i=1}^{N_2}$, then,*

$$\boldsymbol{h}_i^{(2)} = (\Theta_{\overline{\boldsymbol{h}}^{(2)}}^{\boldsymbol{U}'} \circ \Upsilon_{\overline{\boldsymbol{h}}^{(2)}}^{\frac{1}{s}} \circ \Gamma_{\overline{\boldsymbol{h}}^{(1)} \to \overline{\boldsymbol{h}}^{(2)}})(\boldsymbol{h}_i^{(1)}), \ i = 1, \ldots, N_2, \quad (16)$$

*where $\boldsymbol{U}' \in \mathbb{O}(d)$.*

Note that HPA consists of the sequence of Riemannian translation, Riemannian scaling and wrapped-rotation. The domain and range of each component is the manifold $\mathbb{L}^d$. Yet, the first and last operations of each component are the Logarithmic and Exponential maps that project a point from the manifold to the tangent space, and vice versa, respectively. This allows us to propose an efficient implementation of the sequence without the back and forth projections as described in Appendix C.

### 3.5 Extension to multiple sets

We can naturally scale up the setting to support the alignment of $K > 2$ sets, denoted by $\mathcal{H}^{(k)} = \{\boldsymbol{h}_i^{(k)}\}_{i=1}^{N_k}$, where $k \in \{1, 2, \ldots, K\}$. Let $\overline{\boldsymbol{h}}^{(k)}$ and $d^{(k)}$ be the Riemannian mean and dispersion of

---

**Algorithm 1** Hyperbolic Procrustes analysis

---

**Input:** $K$ sets of hyperbolic points $\mathcal{H}^{(1)} = \{\boldsymbol{h}_i^{(1)}\}_{i=1}^{N_1}, \ldots, \mathcal{H}^{(K)} = \{\boldsymbol{h}_i^{(K)}\}_{i=1}^{N_K}$

**Output:** $K$ *aligned* sets of hyperbolic points $\breve{\mathcal{H}}^{(1)} = \{\breve{\boldsymbol{h}}_i^{(1)}\}_{i=1}^{N_1}, \ldots, \breve{\mathcal{H}}^{(K)} = \{\breve{\boldsymbol{h}}_i^{(K)}\}_{i=1}^{N_K}$

1: **for** each set $\mathcal{H}^{(k)}$ **do**
2:    compute the Riemannian mean $\overline{\boldsymbol{h}}^{(k)}$ and dispersion $d^{(k)}$
3: **end for**
4: compute $\overline{\boldsymbol{h}}$, the global Riemannian mean of $\{\overline{\boldsymbol{h}}^{(k)}\}_{k=1}^{K}$
5: **for** each set $\mathcal{H}^{(k)}$ **do**
6:    apply the Riemannian translation $\widetilde{\boldsymbol{h}}_i^{(k)} = \Gamma_{\overline{\boldsymbol{h}}^{(k)} \to \overline{\boldsymbol{h}}}(\boldsymbol{h}_i^{(k)})$              // Eq. (2)
7:    apply the Riemannian scaling $\widehat{\boldsymbol{h}}_i^{(k)} = \Upsilon_{\overline{\boldsymbol{h}}}^{s}(\widetilde{\boldsymbol{h}}_i^{(k)})$ with $s = 1/\sqrt{d^{(k)}}$       // Eq. (8)
8:    apply the wrapped rotation $\breve{\boldsymbol{h}}_i^{(k)} = \Theta_{\overline{\boldsymbol{h}}}^{\boldsymbol{U}}(\widehat{\boldsymbol{h}}_i^{(k)})$ with $\boldsymbol{U} = \boldsymbol{U}^{(1)}(\boldsymbol{U}^{(k)})^{\top}$    // Eq. (11)
9: **end for**

---

the $k$-th dataset, respectively. In addition, let $\overline{\boldsymbol{h}}$ be the global Riemannian mean of $\{\overline{\boldsymbol{h}}^{(k)}\}_{k=1}^{K}$. We propose to transport the points of the $k$-th set using $\Gamma_{\overline{\boldsymbol{h}}^{(k)} \to \overline{\boldsymbol{h}}}$. Next, the Riemannian dispersion of each set is set to 1 by applying $\Upsilon_{\overline{\boldsymbol{h}}}^{s}$ with $s = 1/\sqrt{d^{(k)}}$. Finally, the wrapped rotation is applied to all the data sets on the mapping of the tangent space $\mathcal{T}_{\overline{\boldsymbol{h}}}\mathbb{L}^d$ and then mapped back to the manifold $\mathbb{L}^d$. The first set is designated as the reference set, and all other rotation matrices $\boldsymbol{U}^{(k)}$ are updated according to $\boldsymbol{u}_i^{(k)} \leftarrow \text{sign}(\langle \boldsymbol{u}_i^{(k)}, \boldsymbol{u}_i^{(1)} \rangle) \boldsymbol{u}_i^{(k)}$. The proposed HPA for multiple sets is summarized in Algorithm 1, and some implementation remarks appear in Appendix C.

## 4 Experimental results

We apply HPA to simulations and to three biomedical datasets[1]. In addition, we test HPA on MNIST [31] and USPS [23] datasets, which arguably do not have a distinct hierarchical structure. Nonetheless, we demonstrate in Appendix D that our HPA is highly effective in aligning these two datasets. All the experiments are label-free. We compare the obtained results to the following alignment methods: (i) PAH [51], which is applied only to the simulated data since it requires the existence of a one-to-one correspondence between the sets, (ii) only the Riemannian translation (RT), (iii) OT in hyperbolic space with the weighted Fréchet mean (HOT-F) extended to an unsupervised setting according to [54], (iv) OT with W-linear map (HOT-L) [22], and (v) hyperbolic mapping estimation (HOT-ME) [22]. As a baseline, we present the results obtained before the alignment (Baseline). For more details on the experimental setting, see Appendix C.

### 4.1 Simulations

The synthetic data in $\mathbb{L}^d$ is generated using the sampling scheme described in Section 2 based on [39]. Given an arbitrary point $\boldsymbol{\mu} \in \mathbb{L}^d$ and an arbitrary SPD matrix $\boldsymbol{\Sigma} \in \mathbb{R}^{d \times d}$, we generate a set of $N$ points $\mathcal{Q}^{(1)} = \{\boldsymbol{q}_i^{(1)}\}_{i=1}^{N}$ centered at $\boldsymbol{\mu}$ by $\mathbb{L}^d \ni \boldsymbol{q}_i^{(1)} = \text{Exp}_{\boldsymbol{\mu}}(\text{PT}_{\boldsymbol{\mu}_0 \to \boldsymbol{\mu}}(\widetilde{\boldsymbol{v}}_i^{(1)}))$, where $\boldsymbol{\mu}_0 = [1, \boldsymbol{0}]^{\top}$ is the origin, $\boldsymbol{v}_i^{(1)} = [0, \widetilde{\boldsymbol{v}}_i^{(1)}]^{\top}$, and $\widetilde{\boldsymbol{v}}_i^{(1)} \sim \mathcal{N}(\boldsymbol{0}, \boldsymbol{\Sigma})$. Next, we generate three noisy and distorted versions of $\mathcal{Q}^{(1)}$. The first noisy set $\mathcal{Q}^{(2)} = \{\boldsymbol{q}_i^{(2)}\}_{i=1}^{N}$ is generated as proposed in [51] by $\boldsymbol{q}_i^{(2)} = \boldsymbol{L} \boldsymbol{T}_{\boldsymbol{\epsilon}_i} \boldsymbol{q}_i^{(1)}$, where $\boldsymbol{T}_{\boldsymbol{\epsilon}_i}$ is a hyperbolic translation defined by $\boldsymbol{T}_{\boldsymbol{\epsilon}_i} = [\sqrt{1 + \boldsymbol{\epsilon}_i^{\top} \boldsymbol{\epsilon}_i}, \boldsymbol{\epsilon}_i^{\top}; \boldsymbol{\epsilon}_i, (\boldsymbol{I} + \boldsymbol{\epsilon}_i \boldsymbol{\epsilon}_i^{\top})^{\frac{1}{2}}]$, $\boldsymbol{\epsilon}_i$ is sampled from $\mathcal{N}(\boldsymbol{0}, \sigma^2 \boldsymbol{I})$, $\sigma^2$ is the variance, and $\boldsymbol{L}$ is a random H-unitary matrix [16]. Another noisy set, denoted as $\mathcal{Q}^{(3)} = \{\boldsymbol{q}_i^{(3)}\}_{i=1}^{N}$, is generated by $\boldsymbol{q}_i^{(3)} = \boldsymbol{L}(\text{Exp}_{\boldsymbol{\mu}}(\text{PT}_{\boldsymbol{\mu}_0 \to \boldsymbol{\mu}}(\boldsymbol{u}_i^{(1)})))$, where $\boldsymbol{u}_i^{(1)} = [0, \widetilde{\boldsymbol{v}}_i^{(1)} + \boldsymbol{\epsilon}_i]^{\top}$. Here, the noise is added to the tangent space at $\boldsymbol{\mu}_0$. Finally, let $\mathcal{Q}^{(4)} = \{\boldsymbol{q}_i^{(4)}\}_{i=1}^{N}$ be a distorted set, given by $\boldsymbol{q}_i^{(4)} = f_{\boldsymbol{\mu}'}(\boldsymbol{q}_i^{(3)})$, where $f_{\boldsymbol{\mu}'}(\boldsymbol{x}) = \cosh(||\boldsymbol{u}||_{\mathcal{L}} t) \boldsymbol{\mu}' + \sinh(||\boldsymbol{u}||_{\mathcal{L}} t) \frac{\boldsymbol{u}}{||\boldsymbol{u}||_{\mathcal{L}}}$ and $\boldsymbol{u} = \text{Log}_{\boldsymbol{\mu}'}(\boldsymbol{x})$, for arbitrary (fixed) $\boldsymbol{\mu}' \in \mathbb{L}^d$ and $t > 0$.

---

[1]Our code is available at https://github.com/RonenTalmonLab/HyperbolicProcrustesAnalysis.

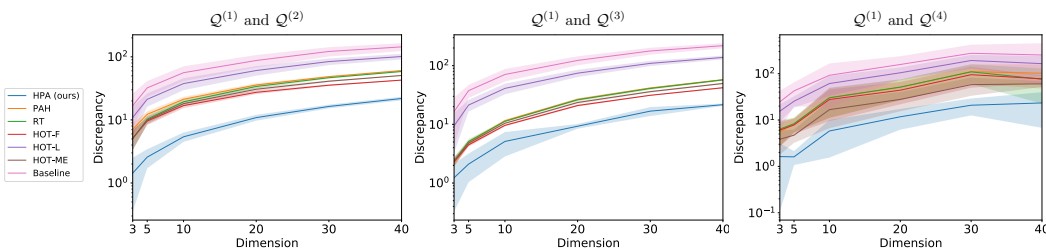

Figure 2: Alignment results of the simulated data.

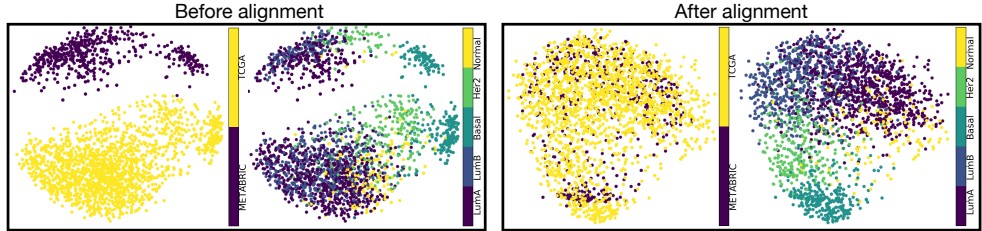

Figure 3: The visualizations of HPA applied to the two breast cancer gene expression datasets.

We apply Algorithm 1 to align the three pairs of sets $\{\mathcal{Q}^{(1)}, \mathcal{Q}^{(2)}\}$, $\{\mathcal{Q}^{(1)}, \mathcal{Q}^{(3)}\}$, and $\{\mathcal{Q}^{(1)}, \mathcal{Q}^{(4)}\}$, setting $N = 100$, $\sigma = 1$, and $d \in \{3, 5, 10, 20, \ldots, 40\}$. Each experiment is repeated 10 times with different values of $\boldsymbol{\mu}$, $\boldsymbol{\Sigma}$, $\boldsymbol{\mu}'$ and $t$. To evaluate the alignment, we use the pairwise discrepancy based on the *hidden* one-to-one correspondence, given by $\varepsilon(\mathcal{Q}^{(1)}, \mathcal{Q}^{(j)}) = \frac{1}{N} \sum_{i=1}^{N} d_{\mathbb{L}^d}^2(\boldsymbol{q}_i^{(1)}, \boldsymbol{q}_i^{(j)})$, where $j \in \{2, 3, 4\}$. The discrepancy as a function of the dimension $d$ is shown in Fig. 2. We observe that the proposed HPA has lower discrepancy relative to the other label-free methods. Specifically, it outperforms OT-based methods that are designed to match the densities. Furthermore, the proposed HPA is stable, in contrast to HOT-L, which is highly sensitive to the noise and distortion introduced in $\mathcal{Q}^{(3)}$ and $\mathcal{Q}^{(4)}$. Interestingly, we remark that the discrepancies of RT and PAH are very close, empirically showing that RT alone is comparable to PAH. In addition, note that HPA is permutation-invariant and does not require one-to-one correspondence as PAH. We report the running-time in Appendix D and demonstrate that HPA is more efficient than HOT-F and HOT-ME.

### 4.2 Batch effect removal

We consider bioinformatics datasets consisting of gene expression data and CyTOF. Representing such data in hyperbolic spaces was shown to be informative and useful [25], implying that such data have an underlying inherent hierarchical structure. Batch effects [43] arise from experimental variations that can be attributed to the measurement device or other environmental factors. Batch correction is typically a critical precursor to any subsequent analysis and processing.

Three batch effect removal tasks are examined. The first task involves breast cancer (BC) gene expression data. We consider two publicly available datasets: METABRIC [8] and TCGA [26], consisting of samples from five breast cancer subtypes. The batch effect stems from different profiling techniques: gene expression microarray and RNA sequencing. In the second task, three cohorts of lung cancer (LC) gene expression data [21] are considered, consisting of samples from three lung cancer subtypes. The data were collected using gene expression microarrays at three different sites (a likely source of batch effects): Stanford University (ST), University of Michigan (UM), and Dana-Farber Cancer Institute (D-F). The last task involves CyTOF data [48] consisting of peripheral blood mononuclear cells (PBMCs) collected from two multiple sclerosis patients during four days: two day before treatment (BT) and two days after treatment (AT). These $8 = 2 \times 2 \times 2$ batches were collected with or without PMA/ionomycin stimulated PBMCs. We aim to remove the batch effects between two different days from the same condition (BT/AT) and from the same patient. In each batch removal task, we first learn an embedding of the data from all the batches into the Lorentz model $\mathbb{L}^d$ [42]. Then, HPA is applied to the embedded points in $\mathbb{L}^d$.

Table 1: The k-NN AUC-ROC from the best k in each method.

| Datasets | Space | S-Baseline | Baseline | HPA | RT | HOT-F | HOT-L | HOT-ME |
|---|---|---|---|---|---|---|---|---|
| BC | $\mathbb{L}^{10}$ | $0.7716 \pm 0.0088$ | $0.5254 \pm 0.0091$ | $\mathbf{0.7410 \pm 0.0354}$ | $0.6001 \pm 0.0658$ | $0.5838 \pm 0.0121$ | $0.4594 \pm 0.0428$ | $0.5600 \pm 0.0147$ |
| LC | $\mathbb{L}^{20}$ | $0.9137 \pm 0.0532$ | $0.5521 \pm 0.0991$ | $\mathbf{0.8316 \pm 0.0904}$ | $0.5318 \pm 0.0370$ | $0.5400 \pm 0.0396$ | $0.5150 \pm 0.0654$ | $0.4901 \pm 0.0516$ |
| P1 BT | $\mathbb{L}^{20}$ | $0.9723 \pm 0.0058$ | $0.6646 \pm 0.1556$ | $\mathbf{0.9401 \pm 0.0068}$ | $0.9020 \pm 0.0086$ | $0.8603 \pm 0.0142$ | $0.8855 \pm 0.0057$ | $0.8919 \pm 0.0089$ |
| P1 AT | $\mathbb{L}^{20}$ | $0.9590 \pm 0.0150$ | $0.7656 \pm 0.1564$ | $\mathbf{0.9329 \pm 0.0011}$ | $0.8270 \pm 0.0880$ | $0.8469 \pm 0.0873$ | $0.8914 \pm 0.0175$ | $0.8848 \pm 0.0306$ |
| P2 BT | $\mathbb{L}^{20}$ | $0.9686 \pm 0.0114$ | $0.6971 \pm 0.1335$ | $\mathbf{0.9329 \pm 0.0186}$ | $0.8830 \pm 0.0142$ | $0.8762 \pm 0.0215$ | $0.8566 \pm 0.1720$ | $0.8810 \pm 0.0040$ |
| P2 AT | $\mathbb{L}^{20}$ | $0.9045 \pm 0.0070$ | $0.5688 \pm 0.0688$ | $\mathbf{0.8453 \pm 0.0798}$ | $0.7190 \pm 0.0439$ | $0.7012 \pm 0.0912$ | $0.7136 \pm 0.0012$ | $0.7291 \pm 0.0113$ |

Table 2: The mean and std of MMD computed over ten random subsets of size 300 for BC, 15 for LC, and 1000 for CyTOF from the two datasets.

| Datasets | Space | Baseline | HPA | RT | HOT-F | HOT-L | HOT-ME |
|---|---|---|---|---|---|---|---|
| BC | $\mathbb{L}^{10}$ | $0.2089 \pm 0.0027$ | $0.0013 \pm 0.0004$ | $0.0072 \pm 0.0011$ | $0.0009 \pm 0.0001$ | $0.0019 \pm 0.0004$ | $\mathbf{0.0007 \pm 0.0002}$ |
| ST&UM | $\mathbb{L}^{20}$ | $0.1072 \pm 0.0051$ | $0.0162 \pm 0.0048$ | $0.0250 \pm 0.0049$ | $0.0023 \pm 0.0002$ | $0.0014 \pm 0.0007$ | $\mathbf{0.0011 \pm 0.0006}$ |
| ST&D-F | $\mathbb{L}^{20}$ | $0.3213 \pm 0.0152$ | $0.0122 \pm 0.0042$ | $0.0150 \pm 0.0106$ | $0.0019 \pm 0.0002$ | $\mathbf{0.0010 \pm 0.0004}$ | $0.0015 \pm 0.0003$ |
| UM&D-M | $\mathbb{L}^{20}$ | $0.0790 \pm 0.0071$ | $0.0168 \pm 0.0090$ | $0.0168 \pm 0.0032$ | $0.0017 \pm 0.0003$ | $\mathbf{0.0012 \pm 0.0006}$ | $0.0029 \pm 0.0005$ |
| P1 BT | $\mathbb{L}^{20}$ | $0.0638 \pm 0.0024$ | $0.0012 \pm 0.0002$ | $0.0020 \pm 0.0002$ | $0.0004 \pm 0.0002$ | $\mathbf{0.0002 \pm 0.0001}$ | $0.0009 \pm 0.0002$ |
| P1 AT | $\mathbb{L}^{20}$ | $0.0598 \pm 0.0014$ | $0.0006 \pm 0.0001$ | $0.0015 \pm 0.0001$ | $0.0003 \pm 0.0001$ | $\mathbf{0.0001 \pm 0.0001}$ | $0.0002 \pm 0.0001$ |
| P2 BT | $\mathbb{L}^{20}$ | $0.0424 \pm 0.0021$ | $0.0012 \pm 0.0001$ | $0.0015 \pm 0.0001$ | $\mathbf{0.0002 \pm 0.0001}$ | $0.0020 \pm 0.0008$ | $0.0009 \pm 0.0003$ |
| P2 AT | $\mathbb{L}^{20}$ | $0.0758 \pm 0.0053$ | $0.0011 \pm 0.0002$ | $0.0013 \pm 0.0002$ | $\mathbf{0.0002 \pm 0.0001}$ | $0.0015 \pm 0.0008$ | $0.0007 \pm 0.0003$ |

Fig. 3 shows a visualization of the embedding of the two breast cancer datasets before and after HPA. For visualization, we project the points in $\mathbb{L}^3$ to the 3D Poincaré ball. Before the alignment, the dominant factor separating between the patients' samples (points) is the batch. In contrast, after the alignment, the batch effect is substantially suppressed (visually) and the factors separating the points are dominated by the cancer subtype.

We evaluate the quality of the alignment in two aspects using objective measures: (i) k-NN classification, with leave-one-batch-out cross-validation, is utilized for assessing the alignment of the intrinsic structure, and (ii) MMD [19] is used for assessing the distribution alignment quality. For the classification, we view the five subtypes of BC, the three subtypes of LC, and the presence of stimulated cells in CyTOF as the labels in the respective tasks. In addition to the results of the different alignment methods, we report the k-NN classification based only on a single batch (*S-Baseline*), which indicates the adequacy of the representation in hyperbolic space to the task at hand.

Table 1 depicts the k-NN classification obtained for the best $k$ per method, and Table 2 shows the MMD. In each task, we set the dimension of the Lorentz model $d$ to the dimension in which the best empirical single-task performance is obtained (S-Baseline). We note that similar results and trends are obtained for various dimensions. Additional results for various $k$ values and an ablation study, showing that the combination of all three components yields the best classification results, are reported in Appendix D.

Although the OT-based methods obtain the best matching between the distributions of the batches, HPA outperforms in all three tasks in terms of classification (see Table 1). In the two gene expression tasks, where the data have multiple labels and we align multiple batches, the advantage of HPA compared to the other methods is particularly significant. In Appendix D, we demonstrate HPA's out-of-sample capabilities on the CyTOF data by learning the batch correction map between the different days from one patient and applying it to the data of the other patient.

### 4.3 Discussion

Alignment methods based on density matching, such as OT-based methods, often overlook an important aspect in purely unsupervised settings. Although sample density is the main data property that can be and need to be aligned, preserving the intrinsic structure/geometry of the sets is important, as it might be tightly related to the hidden labels. Indeed, we see in our experiments that OT-based methods provide a good density alignment (reducing the inter-set variability), as demonstrated by small MMD values (see Table 2). However, the intrinsic structure of the sets (the intra-set variability) is not preserved, as evident by the resulting poor (hidden) label matching, conveyed by the k-NN classification performance (see Table 1). This is also illustrated in the right panel of Fig. 1. There it is visible that the three OT-based methods provide good global alignment of the sets, yet the intrinsic structure is not kept, as implied by the poor color matching.

In contrast to OT-based methods, HPA does not explicitly aim to match densities, and thus, it obtains slightly worse MMD performance compared to OT-based methods. However, HPA matches the first two moments of the density and includes the rotation component, which was shown to be one of the fundamental limitations of OT-based methods for alignment as OT cannot recover volume-preserving maps [4, 36]. As seen in the simulation and experimental results and illustrated in the right panel of Fig. 1, we still obtain a good global alignment and simultaneously preserve the intrinsic structure, allowing for high classification performance. We remark that in the synthetic examples, there is a (hidden) one-to-one correspondence between the sets, and therefore one-to-one discrepancy can be computed (instead or in addition to MMD). When there is such a correspondence, OT still cannot recover volume-preserving maps, while HPA can mitigate noise and distortions.

## 5 Conclusion

We introduced HPA for label-free alignment of data in the Lorentz model. Based on Riemannian geometry, we presented new translation and scaling operations that align the first and second Riemannian moments as well as a wrapped rotation that aligns the orientation in the hyperboloid model. Our theoretical analysis provides further insight and highlights properties that may be useful for practitioners. We empirically showed in simulations that HPA is stable under noise and distortions. Experimental results involving purely unsupervised batch correction of multiple bioinformatics datasets with multiple labels is demonstrated. Beyond alignment and batch effect removal, our method can be viewed as a type of domain adaptation or a precursor of transfer learning that relies on purely geometric considerations, exploiting the geometric structure of data as well as the geometric properties of the space of the data. In addition, it can be utilized for multimodal data fusion and geometric registration of shapes with hierarchical structure.

## Acknowledgments and Disclosure of Funding

We thank the reviewers for their important comments and suggestions, and we thank Thomas Dagès for the helpful discussion. The work of YEL and RT was supported by the European Union's Horizon 2020 research and innovation programme under grant agreement No. 802735-ERC-DIFFOP. The work of YK was supported by the National Institutes of Health R01GM131642, UM1PA05141, P50CA121974, and U01DA053628.

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
