# OpenReview forum: "Hyperbolic Procrustes Analysis Using Riemannian Geometry"
_NeurIPS.cc/2021/Conference — NeurIPS 2021 Poster_

### Official Review · Reviewer_aJ5V · 2021-07-16

**Rating:** 5
**Confidence:** 4

**Summary:**

The authors introduce Hyperbolic Procrustes Analysis (HPA) for unsupervised alignment of hierarchical data in the Lorentz model of hyperbolic space. The proposed method is tested on biomedical data sets, which are assumed to embed well into hyperbolic space due to possessing a hierarchical structure, and additional data sets, which are not assumed to have a hierarchical structure. Some of the experiments indicate improvements over state-of-the-art optimal transport (OT)-based methods.

**Limitations And Societal Impact:**

yes

**Main Review:**

*Assessment*

Overall, this is an interesting paper. My main concerns are about the experimental setup (choice of data sets for assessment) and the comparison with OT-based methods. At present, to me the later does not demonstrate that HPA is superior to OT-based methods. I am open to raising my score if the authors could address these two concerns in the rebuttal.


*Comments*

- Procrustes analysis and the problem of (unsupervised) alignment of data sets is a topical problem and of interest to the ML community. The recent surge of interest in hyperbolic spaces makes the paper relevant.
- Related work:
   -  The paper only sparsely refers to the extensive body of literature on the Procrustes analysis in the field of Riemannian optimization.
  - Consider giving more context on the increasing body of literature on ML in hyperbolic spaces.
  - Recent work [46] seems to have proposed a similar method, albeit with limitations. Please be more specific about the differences between this work and yours. Granted that it seems to be concurrent work, consider adding an experimental comparison.
- While I appreciate the detailed definitions of the tools from Riemannian geometry underlying the proposed method, I do not think that sections 2 and parts of 3 are very accessible to the broader ML community. Consider moving details to the supplemental material and provide a more high-level overview instead. I would illustrate the Lorentz model and the respective geometric tools in a schematic figure.
- Experiments:
  - When using MMD to evaluate the alignment quality the OT-based methods consistently outperform HPA. Can you comment on why this is? It seems to me that the MMD analysis is more general than evaluating classification performance, so I am not convinced that HPA is a better method. If the main argument is computational efficiency, please make this clearer.
  - Hyperbolic ML methods are usually motivated by and demonstrated on data that embeds well into hyperbolic space (vs. Euclidean space) due to a true hierarchical structure. For the data sets analyzed, the authors should provide concrete evidence on good embeddability in hyperbolic space, e.g., by computing embeddings into both hyperbolic and Euclidean spaces and reporting distortion.
  - How does the performance of HPA for data with and without a true hierarchical structure compare? You indicate that it performs well, even on data without a true hierarchical structure. Can you comment on that?
  - How does HPA compare to the Euclidean SOTA for data with and without a true hierarchical structure?


**Time Spent Reviewing:**

---

> ### Author Response · Authors · 2021-08-10
> **Response to Reviewer aJ5V (part 1)**
>
> We thank the reviewer for the careful review and for the important comments.
> We hope that in the following response we were able to address the reviewer's concerns in a satisfactory manner.
>
> 1) **Experimental comparison with OT-based methods:** We thank the reviewer for highlighting this important issue, which was also raised by $\\texttt{reviewer XtjE}$. For convenience, we include our response here as well. We wish to emphasize that we do not argue that our HPA is better than OT-based methods only due to better computational efficiency, but we also argue that it is better in terms of alignment, as explained below.
>
>     One of the overarching messages of our paper, which extends the problem and methodology (alignment in hyperbolic spaces using Riemannian geometry) that were actually being investigated by us here, is that alignment methods based on density matching, such as OT-based methods, often overlook an important aspect. Arguably, in a label-free scenario, sample density is the main data property that can be and need to be aligned. However, by doing that, the intrinsic structure/geometry of the sets might be completely altered. Indeed, we see in our experiments that OT-based methods provide a good density alignment (reducing the inter-set variability), as conveyed by high MMD performance, but the intrinsic structure of the sets (the intra-set variability) is not preserved, as evident by the resulting poor (hidden) label matching, conveyed by the k-nn classification performance. This is also illustrated in the right panel of Fig 1 in the paper. There we visually see that the three OT-based methods provide good global alignment of the sets, yet the intrinsic structure is not kept, as implied by the poor color matching.
>
>     On the other hand, HPA does not explicitly aim to match densities. Therefore, it obtains worse MMD performance compared to OT-based methods. However, HPA matches the first two moments of the density and includes the rotation component, which is completely missing in OT (in fact, this was shown to be one of the fundamental limitations of OT-based methods for alignment as OT cannot recover volume-preserving maps [2,3]). By doing so, as seen in the simulation and experimental results and illustrated in the right panel of Fig 1, we still obtain a good global alignment and simultaneously preserve the intrinsic structure, allowing for high classification performance.
>
>     We remark that in the synthetic examples, there is a (hidden) one-to-one correspondence between the sets, and therefore one-to-one discrepancy can be computed (instead or in addition to MMD). When there is such a correspondence, OT still cannot recover volume-preserving maps, while HPA can mitigate noise and distortions.
>
>     In addition, we remark that our HPA has a natural and stable out-of-sample extension that fits in broader settings, whereas it will be extremely expensive to align new unseen samples in OT-based methods.
>
>     Since our theoretical and experimental results are in the context of hyperbolic spaces, we felt uncomfortable to make such bold statements about the advantage of Procrustes analysis over OT-based methods for label-free data alignment in general in the paper itself. However, we truly appreciate the reviewer's comments on this important issue. We will include the above discussion (with the appropriate modesty) explaining why HPA is considered better than the other tested methods in our revision.
>
> 2) **Choice of datasets and the underlying hierarchical structure:**
>     * In [3], it was shown that gene expression data can be accurately represented in hyperbolic space. Based on similar considerations, we posit that mass cytometry (CyTOF) data exhibit hierarchical structure as well, and as a result, have an informative representation in hyperbolic space. Therefore, we chose to demonstrate our method on gene expression and CyTOF data. We believe that aligning such data is an important and challenging task, and it fits the considered hyperbolic geometry.
>
>     * To empirically test the usefulness of the embedding in hyperbolic space of the specific datasets we considered, we applied a classifier to the embedding of each set separately (marked in the tables by S-Baseline). These classification results do not depend on the proposed alignment method (HPA), but only on the (quality of the) embedding of the data into hyperbolic space and on the ability, using the induced Riemannian distance, to classify the data according to its labels. Indeed, the obtained S-Baseline classification performance is high, indicating that the representation of the data in hyperbolic space is informative and useful.
>
>     * In the appendix, we presented an application of our HPA to the alignment of two digits datasets: MNIST [4] and USPS [5]. In contrast to the bioinformatics data, in this case, there are no definitive latent hierarchical structures. Yet, our HPA still demonstrates good results and outperforms the competing methods. We remark that on the one hand, the reported results do not beat the state-of-the-art. On the other hand, the state-of-the-art performance reported in the literature for the adaptation of these two sets is obtained with (partial) access to labels from both sets, and therefore, comparing it to the results obtained by our label-free method is unfair.
>
>     * As we also stated in the response to $\\texttt{reviewer ywh5}$ under item **Embedding**, we tested classical (Euclidean) Procrustes analysis on the initial embedding of the data, and the obtained results were very poor (we didn't include it in the original submission since we felt that including them would seem as an unfair comparison). We believe that this provides another empirical evidence for the adequacy of the representation in hyperbolic space of the considered data.
>
> 3) **Related work:**
> * **Comparison with PAH [1]:** We thank the reviewer for pointing this important issue, which was also raised by $\\texttt{reviewer EAV6}$. For convenience, we repeat our response here.
>
>      We remark that we empirically compared our work to PAH [1] only in simulations, since PAH requires correspondence between the datasets, which is not available in general, and specifically, in the considered datasets.
>
>     We also wish to remark that PAH [1] was simultaneously submitted to another venue (and later published in [1]), as also acknowledged by reviewer $\\texttt{ywh5}$.
>
>     In the paper, we presented an empirical comparison to PAH. The comparison is done only in simulations, because PAH requires sample correspondence, and the tested datasets do not have sample correspondence.
>
>     Both PAH [1] and our HPA are conceptually based on Procrustes analysis. However the problem setting and the approach are different. Specifically, our HPA makes use of the Riemannian geometry of hyperbolic spaces.
>
>     Following the reviewer's comments, we outline below a comparison on a technical level and emphasize the advantage of our method. We will add such a text to the paper in our revision.
>
>     Importantly, in addition to the following points, we remark that our HPA has a natural and stable out-of-sample extension that fits in broader settings, as demonstrated in the appendix.
>
>     * **Translation:** PAH proposed to translate the centroids of the two sets to the origin in $\mathbb{L}^d$, which is denoted by $\boldsymbol{\mu}_0$. In contrast, we proposed a Riemannian translation of the sets by parallel transport along the unique geodesic path connecting the Fréchet mean of one set to the Fréchet mean of the other set (or from the Fréchet mean of the set to the Fréchet mean of the Fréchet means when multiple sets are aligned). The translations of PAH and HPA are both isometries. However, the translation of our HPA also preserves the geodesic velocities. We also remark that translations cause rotations and distortions. Our translation using PT (derived from the Levi-Civita connection which is torsion-free) introduces the "minimal distortion'' in the Riemannian sense. In addition, the result in Prop. 5 (which is an important property for alignment, allowing for a convenient multi-level alignment) is a direct consequence of the specific translation we propose and does not necessarily hold when the sets are translated to the origin as in PAH.
>     * **Scaling:** PAH assumes that the hyperbolic sets are isometric and it does **not** address the scaling problem at all. In our HPA, the proposed Riemannian scaling is based on the geodesic paths between each point and the Riemannian mean of the set. This way we can align the Riemannian second moment (defined as the dispersion) by taking into account the change of the Riemannian metric as we travel along the geodesic path on the manifold.
>     * **Rotation:** PAH suggested a hyperbolic rotation map that requires one-to-one correspondence. We do not make this assumption, and our Riemannian wrapped rotation component (and the other components) can operate in a broader context. Such an assumption significantly limits the scope of PAH. Indeed, the batch correction applications we considered (nor the standard MNIST and USPS alignment we presented in the appendix) do not have a one-to-one correspondence, and therefore, PAH cannot be tested on these applications. In addition, we note that the hyperbolic rotation map in PAH does not preserve the Riemannian mean if the mean does not coincide with the origin. In PAH, it does not raise a problem since the mean alignment is implemented by translating the sets to the origin. However, in general, and specifically when using the Riemannian translation of our HPA, such a rotation could cancel the mean alignment.

---

> > ### Author Response · Authors · 2021-08-10
> > **Response to Reviewer aJ5V (part 2)**
> >
> > * **Comparisons with Euclidean SOTA:** To the best of our knowledge, in recent work on batch correction of bioinformatics data (with a hierarchical structure) that is considered (the Euclidean) SOTA, e.g., [5,6], the methods are not label-free, and the results are obtained with partial/full access to labels from both the source and target sets. In our work, we consider the (more) challenging scenario of label-free alignment.
> >
> >     Similarly, in the domain adaption of digits datasets, the recent Euclidean SOTA approaches, e.g., [8, 9], address an aligning problem between a **labeled** source dataset and an **unlabeled** target dataset.
> >
> >     We wish to emphasize that in our experimental study, we do not use the labels from the source set nor the labels from the target set. Since the investigated settings are different, we did not report the results of the Euclidean SOTA.
> >
> > * **PA background:** Following the suggestion of the reviewer (which coincides with similar suggestions made by another reviewer), we plan to include a comprehensive literature survey on classical Procrustes analysis (PA) as well as on its Riemannian counterpart in the revision.
> >
> > References
> >
> > [1] P. Tabaghi, and I. Dokmanić. "On Procrustes Analysis in Hyperbolic Space." IEEE Signal Processing Letters 28 (2021): 1120-1124.
> >
> > [2] Y. Brenier. Polar factorization and monotone rearrangement of vector-valued functions. Communications on pure and applied mathematics, 44(4):375–417, 1991.
> >
> >
> > [3] A. Klimovskaia, D. Lopez-Paz, L. Bottou, and M. Nickel. Poincaré maps for analyzing complex hierarchies in single-cell data. Nature communications, 11(1):1–9, 2020.
> >
> >
> > [4] Y. LeCun, L. Bottou, Y. Bengio, and P. Haffner. Gradient-based learning applied to document recognition. Proceedings of the IEEE, 86(11):2278–2324, 1998.
> >
> > [5] J. J. Hull. A database for handwritten text recognition research. IEEE Transactions on pattern analysis and machine intelligence, 16(5):550–554, 1994.
> >
> > [6] I. Korsunsky, N. Millard, J. Fan, K. Slowikowski, F. Zhang, K. Wei, Y. Baglaenko, M. Brenner, P.-r. Loh, and S. Raychaudhuri. Fast, sensitive and accurate integration of single-cell data with harmony. Nature methods, 16(12):1289–1296, 2019.
> >
> > [7] T. Stuart, A. Butler, P. Hoffman, C. Hafemeister, E. Papalexi, W. M. Mauck III, Y. Hao, M. Stoeckius, P. Smibert, and R. Satija. Comprehensive integration of single-cell data. Cell, 177(7):1888–1902, 2019.
> >
> > [8] S. Lee, S. Cho, S. Im. DRANet: Disentangling Representation and Adaptation Networks for Unsupervised Cross-Domain Adaptation. Proceedings of the IEEE/CVF Conference on Computer Vision and Pattern Recognition (CVPR), 2021, pp. 15252-15261.
> >
> > [9] J. Wang, J. Chen, J. Lin, L. Sigal and C.W. de Silva, 2021. Discriminative feature alignment: Improving transferability of unsupervised domain adaptation by Gaussian-guided latent alignment. Pattern Recognition, 116, p.107943.

---

### Official Review · Reviewer_ywh5 · 2021-07-16

**Rating:** 6
**Confidence:** 2

**Summary:**

The paper proposes to adapt Procrust Analysis (PA) to Hyperbolic spaces. Hyperbolic spaces have shown their efficiency for hyperbolic data in previous works. The aim of the work is to use hyperbolic geometry to compute data alignment for hierarchical data. More specifically, the authors derive the three tools used for PA to Lorentz model of hyperbolic space: translation - using parallel transport, logarithmic and exponential map to project to and from the tangent space - , rotation - using a projection to euclidean space - and scaling. Experiments are conducted on artificial datasets in order to evaluate precisely the benefits of the method - knowing the one-to-one correspondence between the two sets to align - and on real datasets.

**Main Review:**

Originality: To the best of my knowledge, this is the first study concerning Procrust Analysis and hyperbolic space - except from [46] cited by the authors and simultaneously submitted to another review (the approach is quite different).

Quality: I have no review all the proofs but according to my understanding the developed techniques and the results seem sound. The protocol setting is judicious, the experiments on artificial data are interesting to assess the quality of the approach and the experiments on real data are convincing. However, from what I understand, the results on real datasets depend on the computed embeddings using [38]. There is no discussion on the effect of this initial embedding of the data and I am curious it is matters or not.

Clarity: The paper is written for specialists of information geometry, it is quite difficult for a non-expert to understand even the purpose of the work. The authors should have introduced better the problematic, at least by introducing PA in the classical setting and discussing more in detail the impact of their work.

Significance: The work is clearly interesting, but the writing of the article - not educational enough - hinders the potential impact of the paper outside of the information geometry community.

**Time Spent Reviewing:**

6

---

> ### Author Response · Authors · 2021-08-10
> **Response to Reviewer ywh5**
>
> We thank the reviewer for the careful and positive review and for the useful feedback.
> We hope that in the following response we were able to address the reviewer's comments in a satisfactory manner.
>
> 1) **Background on classical Procrustes analysis:** Following the reviewer's comment, we realize that background material on classical Procrustes analysis is missing, and we intend to include it in the revision as suggested.
>
> 2) **Embedding:** We wish to remark that our experiments on gene expression data are based on a previous work [1], where it was shown that hyperbolic geometry provides means to discover hierarchies in gene expression datasets.
>
>     In our experiments, we empirically tested the appropriateness of the embedding in (Lorentz) hyperbolic space by applying a classifier to each set separately (marked in the tables by S-Baseline). These results do not depend on the proposed alignment method (HPA), but only on the embedding of the data into hyperbolic space and on the ability, using the induced Riemannian distance, to classify the data according to its labels. Indeed, the obtained S-Baseline classification performance is high, indicating that the representation of the data in hyperbolic space is informative and useful.
>
>     We report that we tested classical (Euclidean) Procrustes analysis on the initial embedding of the data, and the obtained results were very poor, and therefore, we felt that including them in the paper would seem as an unfair comparison. In light of the reviewer's comment, we will present these results in the revision.
>
>     Lastly, we just wish to draw the attention of the reviewer to the fact that the technical details of the embedding we use are outlined in the appendix.
>
> References
>
> [1] A. Klimovskaia, D. Lopez-Paz, L. Bottou, and M. Nickel. Poincaré maps for analyzing complex hierarchies in single-cell data. Nature communications, 11(1):1–9, 2020.

---

### Official Review · Reviewer_EAV6 · 2021-07-19

**Rating:** 6
**Confidence:** 4

**Summary:**

This paper deals with data alignment, and more precisely, of unsupervised alignment of data embedded in hyperbolic spaces. The authors propose a purely geometric approach related to Procrustes analysis (PA), and based on Riemannian geometry. Concerning hyperbolic space, the authors consider the Lorentz model for its numerical efficiency concerning the basic Riemannian calculation.

The hyperbolic PA (HPA) idea is to transpose the classical procrustean analysis to the Lorentzian framework. Hence, the authors generalize the notion of translation using parallel transport, the one of rotation by proposing a "Riemannian wrapped rotation" and finally realizing a geodesic scaling. With these generalizations, the authors can develop their HPA analysis  (which they do).  Theoretical justifications are provided (mainly in supplementary material), as well as numerical illustrations.

**Ethical Concerns:**

I did not identify any ethical issues in this document.

**Limitations And Societal Impact:**

This work is very theoretical. I cannot therefore conclude on the use that could be made of it. As it stands, it is mainly about proposing new tools.


**Main Review:**

The proposed method seems to me innovative and relevant. It is based on strong theoretical arguments (the Lorentzian geometry studied since...) and benefits from a robust algorithmic.

The structure of the article is relevant; the writing of the article is clear and effective. I appreciated the pedagogical effort of the authors to try to bring their method to people not necessarily familiar with Riemannian geometry.

I have several relatively minor questions/comments, which I list below. My main complaint is with the writing in Appendix A. The proofs are awful to follow: successions of calculations without justification, without even a single non-mathematical symbol. We deal more with calculations written in rough drafts than with publishable proofs (especially in a very selective context). I highly recommend a thorough rewrite of these demonstrations.

- The figures are not very legible when printed in black and white. The choice of colors in the legends in Figure 2 and its counterparts in the appendix is not consistent.
- Line 87: Who is $\mu_0$?
- Line 97 about the PAH: In addition to the non-limiting nature, I would have liked to find a brief comparison between the two approaches on a technical level.
- Line 113, about the construction of the Riemannian translation (Equation (2)): A figure illustrating the formula and relating the different objects to each other would be a valuable aid to intuition.
- Line 163 (and following): The tangent space is topologically equivalent to $\mathbb{R}^d$; talking about projection seems to me very improper here.
- Line 117: $\hat{\mathcal{H}}^{(2)}$ is not defined.
- Equation (12): Please improve the parenthesis (different sizes/shapes) to make this equation more readable.
- Line 207: Is there a relationship between $U'$ and $U$?
- The authors mention the hierarchical structure several times (for example, in the last sentence of the article). I would like to understand what they mean by this: the connection with the Lorentzian manifold is unclear to me, unless I don't have the same meaning as the authors.

**Time Spent Reviewing:**

5

---

> ### Author Response · Authors · 2021-08-10
> **Response to Reviewer EAV6**
>
> We thank the reviewer for the careful and positive review, and for the useful feedback and suggestions. We hope that in the following response we were able to address the reviewer's questions in a satisfactory manner.
>
> 1) **Proofs:** We accept and appreciate the reviewer's criticism. As we believe the theoretical and technical aspects of the paper are important, we will improve the presentation of the proofs and make them easier to read and follow, as suggested by adding text, explanations, and justifications in our revision.
>
> 2) **Comparison with PAH [1] (line 97):** First, let us remark that [1] was simultaneously submitted to another venue (and later published in [1]), as also acknowledged by $\\texttt{reviewers ywh5}$ and $\\texttt{aJ5V}$.
>
>     In the paper, we presented an empirical comparison to PAH. The comparison of our HPA to PAH was done only in simulations because PAH requires sample correspondence, and the tested datasets do not have sample correspondence.
>
>     Please note that both PAH [1] and our HPA are conceptually based on Procrustes analysis. However, the problem setting and the approach are different. Specifically, our HPA makes use of the Riemannian geometry of hyperbolic spaces.
>
>     Following the reviewer's comment, we outline below a comparison on a technical level and emphasize the advantage of our method. We will add such text to the paper in our revision.
>
>     Importantly, in addition to the following points, we remark that our HPA has a natural and stable out-of-sample extension that fits in broader settings, as demonstrated in the appendix.
>
>     * **Translation:** PAH proposed to translate the centroids of the two sets to the origin in $\mathbb{L}^d$, which is denoted by $\boldsymbol{\mu}_0$. In contrast, we proposed a Riemannian translation of the sets by parallel transport along the unique geodesic path connecting the Fréchet mean of one set to the Fréchet mean of the other set (or from the Fréchet mean of the set to the Fréchet mean of the Fréchet means when multiple sets are aligned). The translations of PAH and HPA are both isometries. However, the translation of our HPA also preserves the geodesic velocities. We also remark that translations cause rotations and distortions. Our translation using PT (derived from the Levi-Civita connection which is torsion-free) introduces the "minimal distortion'' in the Riemannian sense. In addition, the result in Prop. 5 (which is an important property for alignment, allowing for a convenient multi-level alignment) is a direct consequence of the specific translation we propose and does not necessarily hold when the sets are translated to the origin as in PAH.
>
>     * **Scaling:** PAH assumes that the hyperbolic sets are isometric and it does **not** address the scaling problem at all. In our HPA, the proposed Riemannian scaling is based on the geodesic paths between each point and the Riemannian mean of the set. This way we can align the Riemannian second moment (defined as the dispersion) by taking into account the change of the Riemannian metric as we travel along the geodesic path on the manifold.
>
>     * **Rotation:** PAH suggested a hyperbolic rotation map that requires one-to-one correspondence. We do not make this assumption, and our Riemannian wrapped rotation component (and the other components) can operate in a broader context. Such an assumption significantly limits the scope of PAH. Indeed, the batch correction applications we considered (nor the standard MNIST and USPS alignment we presented in the appendix) do not have a one-to-one correspondence, and therefore, PAH cannot be tested on these applications. In addition, we note that the hyperbolic rotation map in PAH does not preserve the Riemannian mean if the mean does not coincide with the origin. In PAH, it does not raise a problem since the mean alignment is implemented by translating the sets to the origin. However, in general, and specifically when using the Riemannian translation of our HPA, such a rotation could cancel the mean alignment.
>
> 3) **Terminology in the wrapped rotation component (line 163 and following):**  We thank the reviewer for this comment. We agree with the reviewer's comment that the term "projection'' may not be appropriate here. In the revision, we will use the term *mapping* instead.
>
> 4) **Prop. 10 (line 207):** n Prop. 10, we showed that if the discrepancy between the two sets is derived by translation, scaling, and rotation, then our method can perfectly align the sets. However, these operations do not jointly commute. Therefore, the order of the operations is important and perfect alignment is obtained using a possibly different rotation. Namely, $\mathbf{U}$ and $\mathbf{U}'$ are not directly linked (because the two rotations are applied with respect to different reference points: $\overline{\mathbf{h}}^{(1)}$ and $\overline{\mathbf{h}}^{(2)}$).
>
> 5) **Hierarchical structure:** By hierarchical structure, we refer to data with underlying *tree-structure* that, with the addition of an appropriate similarly measure, can be accurately embedded in a hyperbolic space [2,3], where geodesic paths exponentially grow (analogously to climbing up the levels of a tree). Specifically, the Lorentz model (hyperboloid model) is one of the common models of hyperbolic spaces that can be viewed as a continuous counterpart of (discrete) tree structures [3]. We will make this point clear in the revision.
>
> 6) **Presentation:** We thank the reviewer for the constructive suggestions. They will be taken into account in the revision:
>     * Eq. 12: we will change the size and shape of the parentheses.
>     * We will modify the colors of the figures to fit black-and-white printing and correct the consistency with the figures in the appendix.
>     * We will add an illustrated figure of the Riemannian translation (describing Eq. (2)).
>
> 7) **Notation:**
>     *  Line 87: $\mathbf{\mu}_0$ is the origin of $\mathbb{L}^d$. It is defined in line 69, but we will repeat it here.
>     *  $\widehat{\mathcal{H}}^{(2)}$ is defined in Prop. 6 (line 158).
>
> References
>
> [1] P. Tabaghi, and I. Dokmanić. "On Procrustes Analysis in Hyperbolic Space." IEEE Signal Processing Letters 28 (2021): 1120-1124.
>
> [2] M. Nickel and D. Kiela. Poincaré embeddings for learning hierarchical representations. In Advances in Neural Information Processing Systems (NIPS), page 6341–6350, 2017.
>
> [3] M. Nickel and D. Kiela. Learning continuous hierarchies in the Lorentz model of hyperbolic geometry. In International Conference on Machine Learning (ICML), pages 3779–3788. PMLR, 2018.

---

### Official Review · Reviewer_XtJE · 2021-07-19

**Rating:** 7
**Confidence:** 4

**Summary:**

In this paper, the authors rebuild the three components of Procrustes analysis (i.e. translation, scaling, and rotation) using the Riemannian geometry of the Lorentz model, thus practically reimplementing PA for hyperbolic spaces (HPA). Their methodology follows a theoretical analysis which, as the authors state, has its own merit. Furthermore, experiments show how the approach, in addition to being computationally more efficient than previous ones, also is able to better tackle the problem of "batch effect" on hierarchical datasets.

**Ethical Concerns:**

To my understanding, the paper does not raise particular ethical concerns.
The bioinformatic datasets used are public and the data is completely anonymised, without information which would make the original subjects identifiable.

**Limitations And Societal Impact:**

The authors properly addressed the method's limitations. No societal impacts have been considered.

**Main Review:**

Originality:
The presented method looks novel to the extent of my knowledge. Some of the methods employed to derive the new HPA components are not new (e.g. Chami et Al. "Hyperbolic Graph Convolutional Neural Networks" shows how to employ the composition of Exp and Log maps to allow operators which cannot work directly in the hyperbolic space), however here they are used differently and better grounded into theory.

Quality:
The submission looks theoretically sound to me: the method is both theoretically justified and all the steps are properly derived.

Experiments have been conducted to compare HPA's alignment with other methods on increasingly misaligned datasets (with synthetic data); to qualitatively show how it can help reduce the batch effect; and, finally, to quantitatively compare its performances wrt other methods both in terms of k-NN classification and of maximum mean discrepancy.
It is not clear to me why HPA has a worse MMD performance compared to other methods, considering how better it performed both in the synthetic examples case and in the classification task (i.e. the same data where it had bigger discrepancy). I think the authors should provide some reasoning about why this happens and why their approach can still be considered better than others.

Clarity:
I think the paper is well written and clear enough to be replicated. Proofs are made available in Appendix and code is provided with supplementary material for replication.

Significance:
The presented approach has a value per se, regardless of the very specific application shown, in all those cases where the data representation, whether provided or learned, has some intrinsic hierarchical nature and different modalities. The experiments in the paper are rather limited in nature, but in my opinion they serve the purpose of showing the method's quality and making it available to the wider research community.

**Time Spent Reviewing:**

5

---

> ### Author Response · Authors · 2021-08-10
> **Response to Reviewer XtJE**
>
> We thank the reviewer for the careful and positive review, and for the useful feedback.
> We hope that in the following response we were able to address the reviewer's concern regarding the quantitative evaluation of the experimental results in a satisfactory manner.
>
> 1) **Quantitative evaluation of the experimental results:** One of the overarching messages of our paper, which extends the problem and methodology (alignment in hyperbolic spaces using Riemannian geometry) that were actually being investigated by us here, is that alignment methods based on density matching, such as OT-based methods, often overlook an important aspect. Arguably, in a label-free scenario, sample density is the main data property that can be and need to be aligned. However, by doing that, the intrinsic structure/geometry of the sets might be completely altered. Indeed, we see in our experiments that OT-based methods provide a good density alignment (reducing the inter-set variability), as conveyed by high MMD performance, but the intrinsic structure of the sets (the intra-set variability) is not preserved, as evident by the resulting poor (hidden) label matching, conveyed by the k-nn classification performance. This is also illustrated in the right panel of Fig 1 in the paper. There we visually see that the three OT-based methods provide good global alignment of the sets, yet the intrinsic structure is not kept, as implied by the poor color matching.
>
>     On the other hand, HPA does not explicitly aim to match densities. Therefore, it obtains worse MMD performance compared to OT-based methods. However, HPA matches the first two moments of the density and includes the rotation component, which is completely missing in OT (in fact, this was shown to be one of the fundamental limitations of OT-based methods for alignment as OT cannot recover volume-preserving maps [2,3]). By doing so, as seen in the simulation and experimental results and illustrated in the right panel of Fig 1, we still obtain a good global alignment and simultaneously preserve the intrinsic structure, allowing for high classification performance.
>
>     We remark that in the synthetic examples, there is a (hidden) one-to-one correspondence between the sets, and therefore one-to-one discrepancy can be computed (instead or in addition to MMD). When there is such a correspondence, OT still cannot recover volume-preserving maps, while HPA can mitigate noise and distortions.
>
>      Since our theoretical and experimental results are in the context of hyperbolic spaces, we felt uncomfortable to make such bold statements about the advantage of Procrustes analysis over OT-based methods for label-free data alignment in general in the paper itself. However, we truly appreciate the reviewer's comments on this important issue. We will include the above discussion (with the appropriate modesty) explaining why HPA is considered better than the other tested methods in our revision.
>
> 2) **Extent of experiments:** We just wish to draw the attention of the reviewer to the additional experiments we included in the appendix. In addition to the batch correction experiment presented in the paper (involving gene expression alignment and mass cytometry), in the appendix we also demonstrate the out-of-sample-extension capabilities on mass cytometry data, and present alignment results on the standard MNIST and USPS (although these data might not have obvious hierarchical structure).
>
> 3) **Related work:** We thank the reviewer for directing our attention to [1]. We will cite and refer to it in our revision.
>
> References
>
> [1] I. Chami, Z. Ying, C. Ré, and Jure Leskovec. Hyperbolic graph convolutional neural networks.
> In Advances in Neural Information Processing Systems, pages 4869–4880, 2019.
>
> [2] Y. Brenier. Polar factorization and monotone rearrangement of vector-valued functions. Communications on pure and applied mathematics, 44(4):375–417, 1991.
>
> [3] R. J. McCann. Polar factorization of maps on Riemannian manifolds. Geometric and Functional Analysis, 11(3):589-608 ,2001.

---

> > ### Comment · Reviewer_XtJE · 2021-09-02
> > **Reply to authors**
> >
> > Dear authors,
> >
> > thank you very much for your reply and the clarification about my main concern (better KNN performance but worse MMD).
> > I also read my colleagues' reviews and I realised that I oversaw some clarity issues due to the fact I already had some knowledge about PA. As a suggestion, I think having a small "background on PA" as well as the one on hyperbolic geometry you already introduced, to clarify the problem and introduce its basic formulation -the one you then extend in the hyperbolic space-, would definitely improve the paper's clarity for readers who have not previously heard about PA, allowing your work to have a broader impact.

---

> > > ### Author Response · Authors · 2021-09-02
> > > **Response to Reviewer XtJE**
> > >
> > > We would like to thank the reviewer for the thoughtful comments and efforts towards improving our paper.
> > > As suggested, we will include some background on PA in our revision.

---

### Decision · Program_Chairs · 2021-09-28

**Decision:**

Accept (Poster)

**Comment:**

This was a very borderline paper. The original submission appears to be technically sound and novel, but reading the paper without difficulty requires a fair amount of prior expertise in PA. It is dense to the point that it might impact the significance of the work in the general NeurIPS community. However, the reviewers greatly appreciated the enthusiastic interaction with the authors during the rebuttal, and have generally agreed to accept the paper under the assumption that the authors make significant improvements to clarity in the camera ready. As just one example, the reviewers strongly recommend including a high-level introduction of PA in the final submission. Please read through all of the reviewer feedback carefully and follow through on promised adjustments.

**Consistency Experiment:**

NeurIPS has a long history of experimentation. In 2014, NeurIPS ran an experiment in which 10% of submissions were reviewed by two independent committees to quantify the randomness in the review process. This year, we repeated a variant of this experiment to see how the quality of the review process has changed over time.  This paper was part of the experiment and was therefore assigned to two committees (consisting of reviewers, an Area Chair, and a Senior Area Chair) that reached independent decisions.  If both committees made the same recommendation, this recommendation was followed. If a single committee recommended acceptance, the paper was accepted (with the exception of a few cases in which the other committee identified what we considered a fatal flaw, e.g., an error in a key result).

Both committees reached the same decision: **Accept (Poster)**

The other committee assigned to the paper recommended **Accept (Poster)**.  You can find the other set of reviews, along with any follow up discussion with the authors here:
https://openreview.net/forum?id=KWxdXBo-7eNVL